# Massive and parallel expression profiling using microarrayed single-cell sequencing

Sanja Vickovic[1], Patrik L. Ståhl[2,*], Fredrik Salmén[1,*], Sarantis Giatrellis[2], Jakub Orzechowski Westholm[3], Annelie Mollbrink[4], José Fernández Navarro[2], Joaquin Custodio[4], Magda Bienko[4], Lesley-Ann Sutton[5], Richard Rosenquist[5], Jonas Frisén[2] & Joakim Lundeberg[1]

Single-cell transcriptome analysis overcomes problems inherently associated with averaging gene expression measurements in bulk analysis. However, single-cell analysis is currently challenging in terms of cost, throughput and robustness. Here, we present a method enabling massive microarray-based barcoding of expression patterns in single cells, termed MASC-seq. This technology enables both imaging and high-throughput single-cell analysis, characterizing thousands of single-cell transcriptomes per day at a low cost (0.13 USD/cell), which is two orders of magnitude less than commercially available systems. Our novel approach provides data in a rapid and simple way. Therefore, MASC-seq has the potential to accelerate the study of subtle clonal dynamics and help provide critical insights into disease development and other biological processes.

[1] Science for Life Laboratory, Division of Gene Technology, School of Biotechnology, KTH Royal Institute of Technology, SE-114 28 Stockholm, Sweden. [2] Department of Cell and Molecular Biology, Karolinska Institute, SE-171 77 Stockholm, Sweden. [3] Science for Life Laboratory, Department of Biochemistry and Biophysics, Stockholm University, SE-106 91 Stockholm, Sweden. [4] Science for Life Laboratory, Department of Medical Biochemistry and Biophysics, Karolinska Institute, SE-171 77 Stockholm, Sweden. [5] Science for Life Laboratory, Department of Immunology, Genetics and Pathology, Uppsala University, SE-751 85 Uppsala, Sweden. * These authors contributed equally to this work. Correspondence and requests for materials should be addressed to J.L. (email: joakim.lundeberg@scilifelab.se).

RNA sequencing has been an invaluable tool for gene expression analysis[1] that has recently progressed from bulk analysis and averaging multiple cells' transcriptome profiles to single-cell profiling. We have advanced from studying group-specific or condition-dependent fold-changes using microarrays[2] to transcript counting[3] and isoform analysis[4]. This has afforded the potential to unravel both variations among individual cells and stochastic changes across the gene body[5].

Averaging gene expression levels in a population of cells is beneficial when comparing states of particular tissues in different conditions or developmental stages, and this approach has provided numerous advances and biomarkers for diverse pathological, and other conditions[6]. However, it cannot clarify the discrete roles of individual cells nor the transcriptomic triggers responsible for changes in their phenotypes[7]. In addition, scarcity of biological material often precludes the profiling of rare cell populations by conventional RNA sequencing methods[8].

There have been major recent technological breakthroughs[9–12] in the ability to analyse single cells, using methods including cell encapsulation in droplets[13,14], solid-surface complementarity DNA (cDNA) analysis[15,16] and in situ messenger RNA (mRNA) hybridizations[17]. These methods enable quantitative analysis of gene expression in single cells[18] and have been applied, for example, to study of mouse embryogenesis[19] and expression bimodality[20]. Nevertheless, these methods do not provide any possibilities in combining cell imaging and transcriptome profiling, exhibit low-throughput by analysing a single cell at a time or require expensive droplet instrumentation when available at high-throughput.

In this paper, we describe a novel method, termed micro-arrayed single-cell sequencing (MASC-seq), a single tube approach for analysis of single cells using a barcoded microarray, and demonstrate its ability to profile single cells, in both model cell lines and primary chronic lymphocytic leukaemia (CLL) patient cells. MASC-seq can both image cells to provide qualitative information on cells' morphology and profile the expression of hundreds to thousands of single cells daily, far more than current standard procedures based on fluorescence-activated cell sorting (FACS) into plates or single-cell picking into individual reaction volumes[10]. MASC-seq could be compared to commercially available systems such as the Fluidigm C1 (ref. 21), which also provides an imaging system before library preparation. However, MASC-seq is improved in terms of daily throughput, not limited by cell size and also is the first system that enables cDNA synthesis of single cells to run in parallel in a single-reaction lowering chances of technical variation in library preparation. MASC-seq is based on commercially available products and reagents and requires only an extra imaging system when compared with standard RNA-sequencing.

## Results

**Principles of MASC-seq technology.** With MASC-seq, single cells can either simply be smeared and randomly positioned or FACS sorted onto a $6.5 \times 6.8$ mm$^2$ microarray of barcoded DNA oligonucleotides printed in a $33 \times 35$ matrix with 200 µm centre-to-centre pitch (Fig. 1). The matrix contains 1,007 unique DNA barcodes surrounded by a frame used for orientation during positioning. After attachment, a high-resolution image is taken, which links the position of each barcode sequence with each individual cell, and provides information concerning cell morphology. The image also gives information about the number of cells present on top of each barcoded oligonucleotide spot. In MASC-seq the cDNA is synthesized in a hybridization cassette from ∼500 single (given 47% occupancy) cells simultaneously in a single well, thereby reducing possibilities of technical variation in the single-cell cDNA synthesis and library preparation steps. This not only increases robustness, but also lowers time and labour costs. After cDNA synthesis, the cells are removed from the microarray surface by proteinase K digestion and the probes are cleaved from the surface with a uracil-specific excision reagent enzyme, which targets the uracil sequence located at the 5′ end of the microarray barcodes. Each cell barcode consists of a uniquely designed 18 nt sequence[22] followed by a unique molecular identifier (UMI), for individual transcript counting, and an oligo-dTVN sequence, thus the method involves 3′ tagging (Fig. 1). The cleaved material is ready for in vitro amplification[23] and library preparation following a procedure similar to the cell expression by linear amplification and sequencing (CEL-seq) protocol[11]. Around 10,000 single-cell libraries can be prepared, for subsequent sequencing, in 2 days.

**Human adenocarcinoma cell line as a model system.** A human breast adenocarcinoma cell line, MCF-7, was used as a model system to evaluate the quality and quantity of data produced using the MASC-seq method. An experiment was first performed to establish where the cDNA was labelled during the reverse transcription reaction (Fig. 2a). This generated a high-resolution fluorescent (Cy3) print, which could be superimposed with the image of either haematoxylin-stained (Fib. 2b) or fluorescently labelled cells taken before cDNA synthesis, providing a convenient way to assess the cells' quality and visually colocalize the

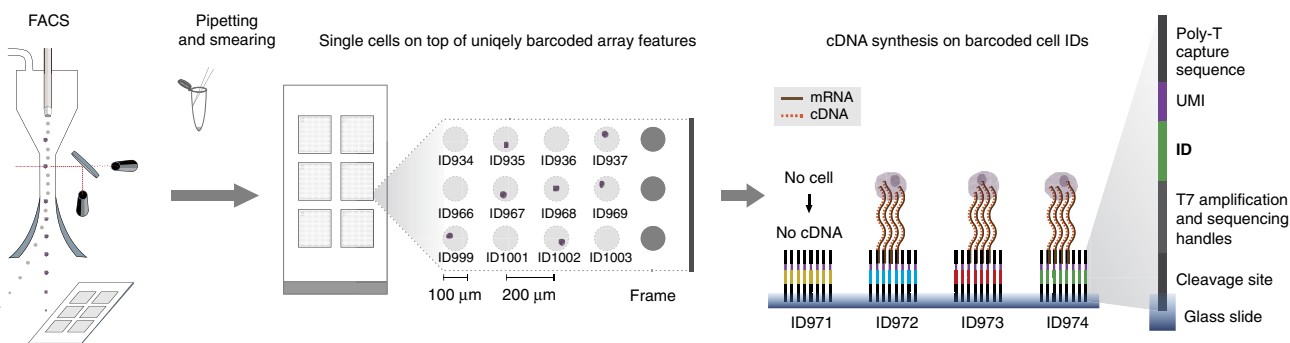

**Figure 1 | MASC-Seq overview.** A FACS machine sorts single cells onto a barcoded microarray, printed with six replicates on an activated glass slide. The throughput of the method and microarray design as a $33 \times 35$ ID matrix is illustrated. An alternative is to pipette and smear cells which then distribute randomly onto the array. Positions of the cells and IDs are noted in a high-resolution image and cDNA is only transcribed when an individual cell lands on top of the barcoded oligo-dTVN primer (ID).

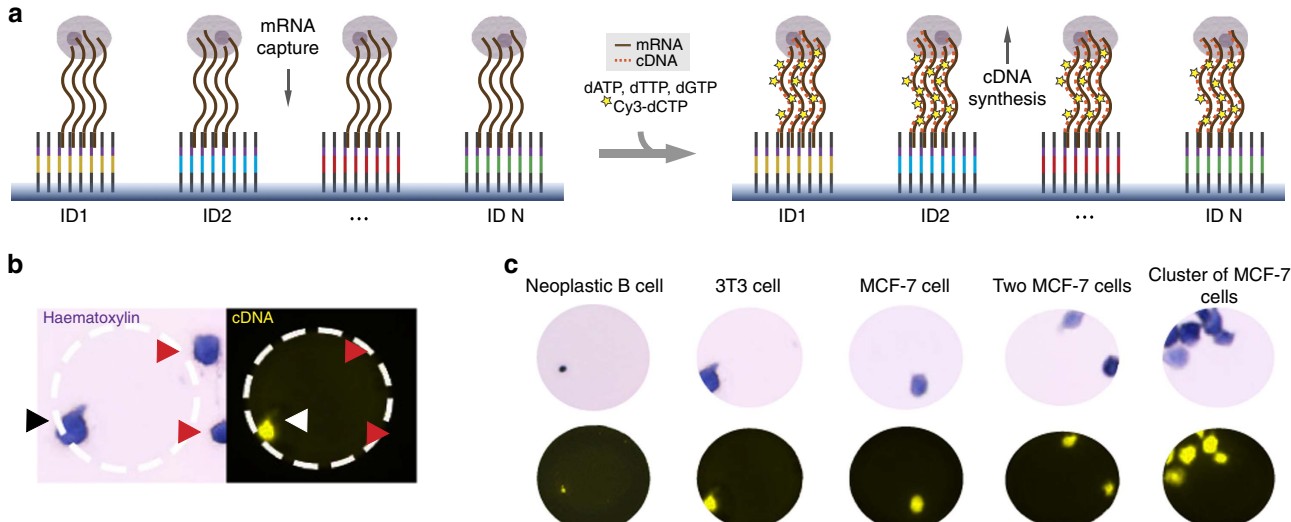

**Figure 2 | Schematic illustration and results of cDNA and cell colocalization.** (**a**) cDNA is labelled with Cy3 nucleotides during reverse transcription. (**b**) A haematoxylin image is taken before cDNA synthesis, and a Cy3 image after cDNA synthesis and removal of cells from the microarray. Cell (black arrow) and cDNA (white arrow) can then be colocalized given the cell is positioned on top of the ID primers (inner circle of the white dashed area). Cells not positioned on top of ID primers do not undergo reverse transcription (red arrows). (**c**) Cell-cDNA colocalization prints for cells used in the study.

cDNAs with single cells. This colocalization confirmed that each cell would produce decoded reads with the correct cell-barcode combination (Fig. 2b, black and white arrow). Furthermore, the cDNA print could only be created when cells were positioned on top of the cell barcode containing the oligo-dTVN sequence (Fig. 2b, red arrows). The results were visually inspected for all cell types used in the study (Fig. 2c). Diffusion of the cDNA signal from the cells' borders was estimated at $0.81 \pm 1.46\,\mu m$ for MCF-7 and at $0.86 \pm 1.96\,\mu m$ for 3T3 cells (Supplementary Fig. 1d,e).

We smeared cells onto the barcoded array, creating four groups of barcoded libraries: singles (1 cell), doublets (2 cells), clusters ($>$2 cells) and background (0 cells). The groups could be decoded based on the positional information from the high-resolution image.

A total of six libraries were prepared and three libraries were shared per slide and operator to evaluate reproducibility and robustness of the protocol. On average 72% of the sequence reads for each library mapped to the reference transcriptome with around 20,000 unique protein-coding genes expressed per library (Fig. 3a) and 82% saturation of unique transcripts at $\sim$384,000 raw reads per barcode (Fig. 3b). Furthermore, $>$94% of the barcodes could be correctly demultiplexed and assigned to a specific barcode group. At this sequencing depth, the single-cell libraries yielded on average 27,427 unique transcripts and 6,293 unique protein-coding genes per cell, with the doublet and cluster libraries following a similar pattern (Fig. 3c).

Irrespective of the number of cells per barcode, cell libraries clearly separated from the background, which generated the fewest reads and the fewest unique protein-coding genes. Furthermore, the background libraries, likely to represent cell-free RNA from lysed cells present in the collection buffer before attachment to the array, yielded higher mean coefficients of variation of gene expression levels, regardless of their strength of expression, while the cell libraries all exhibited very similar profiles, in which signal dispersion was negatively correlated with expression levels (as expected)[24] (Supplementary Fig. 2a).

We also confirmed that UMIs are important for reducing noise, as indicated by a lowered mean coefficient of variation (CV) over gene expression after UMI filtering (Supplementary Fig. 2b)[9]. In addition, we assessed similarities between the groups using Pearson's correlation coefficients and visualized the results

with t-statistic Stochastic Neighbor Embedding (t-SNE)[25] (Fig. 3d) – the background again successfully separated from cell libraries whilst the cell libraries all formed a single cluster, irrespective of the array, slide or operator (Supplementary Fig. 2c,d).

More than 99% of the genes in the single-cell libraries were also found in the bulk RNA sequencing experiments (Fig. 3e). Two types of bulk RNA sequencing libraries were obtained, by polyA-selected cDNA synthesis from 300 ng total RNA via reverse transcription either in solution or anchored on the barcoded microarray surface. In both cases, the single-cell average ($n = 136$) correlated well (Fig. 3f and Supplementary Fig. 3a) with data obtained from the bulk population-average experiments ($R = 0.92$ and $R = 0.90$, respectively). A total 15,937 genes were identified in the single-cell and bulk experiments, of which 1,970 were found only in either of the bulk samples, resulting in an average dropout rate (DOR) of 3.4–6.6% in the single-cell sample (Fig. 3f and Supplementary Fig. 3a). Notably the bulk libraries also had DORs of 3.3 and 7.4% compared with each other.

Additionally, we compared the acquired gene expression profiles of the single cells between each other (Supplementary Fig. 3b,c) and noted that the expression profiles can vary significantly between single cells (DOR $= 35.46\%$, $n = 136$). In summary, these findings demonstrate the quality and reproducibility of the data obtained using our MASC-seq technique.

**Efficiency and sensitivity**. MASC-seq efficiency as well as variation in MCF-7 cells were compared with single FACS sorted MCF-7 cells prepared with the CEL-Seq protocol[11]. To assure a fair comparison, cells were taken from the same culturing plate on the same day for both experiments. Also, a UMI-based primer was used when preparing libraries with the CEL-Seq approach to ensure possibility for UMI filtering. Randomly selected cells from the MASC-seq protocol and the same number of cells created with the CEL-Seq protocol ($n = 36$ for each) shared 64% of the genes (Fig. 4a), with MASC-seq detecting 1.4 times more protein-coding genes in total. On average, MASC-seq captures 17.5 times more unique transcripts and 6.5 times unique protein-coding genes per cell (Fig. 4b). However, most importantly, MASC-seq achieves lower variation of gene expression between cells, as

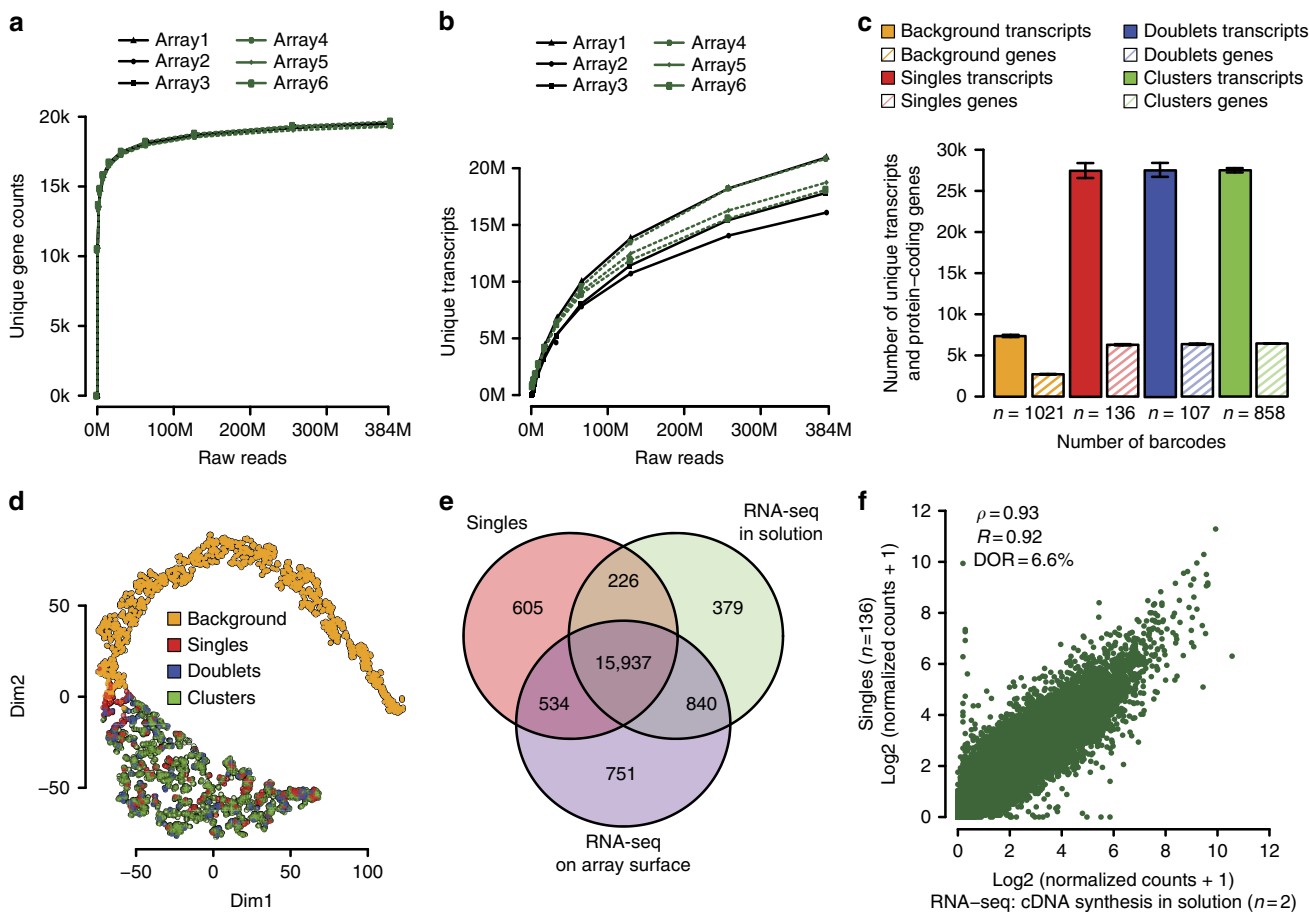

**Figure 3 | MCF-7 as a model system.** (**a**) Number of unique protein-coding genes annotated at a certain sequencing depth for all arrays. Number of the *y* axis are a sum for all barcodes found in the library. Arrays 1–3 were processed on one slide by the first operator, while arrays 4–6 were processed on another slide by a different operator. (**b**) Number of unique transcripts annotated at a certain sequencing depth. (**c**) Numbers of unique protein-coding genes and transcripts found in each barcode group: background (no cells), singles (one cell), doublets (two cells) and clusters ($>2$ cells). Error bars represent s.e.m. (**d**) *t*-SNE plot for all barcode groups: background (no cells), singles (one cell), doublets (two cells) and clusters ($>2$ cells). (**e**) Venn diagram depicting numbers of protein-coding genes shared by single-cell libraries ($n=136$) and libraries obtained from standard RNA-sequencing experiments, in which cDNA was synthesised either in solution ($n=2$) or on the barcoded microarray surface ($n=4$). (**f**) Correlation plot between the single-cell average and RNA-sequencing average (with cDNA synthesis performed in solution) data.

compared with CEL-Seq, due to a single-tube reaction principle (Fig. 4c). Furthermore, single-molecule fluorescent *in situ* hybridization (smFISH) was performed on MCF-7 cells to determine the sensitivity. Absolute numbers of transcripts for seven well characterized genes present in a single cell were compared between the platforms. The sensitivity for the MASC-seq technique was determined to 17.3% (Fig. 4d).

**Barcode crosstalk**. To estimate the degree of barcode crosstalk (to ensure that the data decoded for each cell were accurate and contained within the corresponding cell-barcode combination), we mixed human MCF-7 and mouse 3T3 fibroblasts and smeared them on the barcoded microarray. We then estimated species-specific transcript counts for each of the barcodes that had received a single cell (Supplementary Fig. 4a) based on the high-resolution image. In the single-cell libraries, which produced over 2,000 reads and revealed 1,000 uniquely expressed protein-coding genes per cell (Supplementary Fig. 4b,c), only 30–40 reads were misassigned per barcode, thus only 1.42–1.68% of the total species-specific reads were misassigned to human and mouse barcodes, respectively. Furthermore, *t*-SNE separated species even at the orthologous gene level (Supplementary Fig. 4d), generating two distinct clusters with all cells from each species clustered

together. Population averages correlated well with those of pure samples ($R>0.88$; Supplementary Fig. 4e,f), further confirming the quality of the data.

**Differential expression in single leukaemic cells**. To assess the applicability of our method for studying a disease state, we analysed primary single-sorted neoplastic B cells obtained from three patients diagnosed with CLL, assigned to different major CLL subsets, with distinct clinical and biological characteristics (clinically classified poor-prognostic subsets #1 and #2, and the good-prognostic subset #4)[26,27]. CD5$^+$CD19$^+$ cells from patients were FACS sorted onto MASC-seq arrays ($n=1,189\pm186$ cells per case). As expected, these small-sized cells (7–10 μm) showed a proportionally lower amount of labelled cDNA in each cell (Fig. 2c, left panel).

Average gene expression levels in cells of the three CLL subsets clearly differed (Supplementary Fig. 5a), with only 43% of the expressed genes shared (Supplementary Fig. 5b), and the strongest differences between the subsets were among the most abundantly expressed genes (Supplementary Fig. 5c). Notably, comparing the 500 genes with highest expression in each subset, only 30 genes showed high expression in two subset pairs and no genes were shared in all three subsets (Supplementary Fig. 5d).

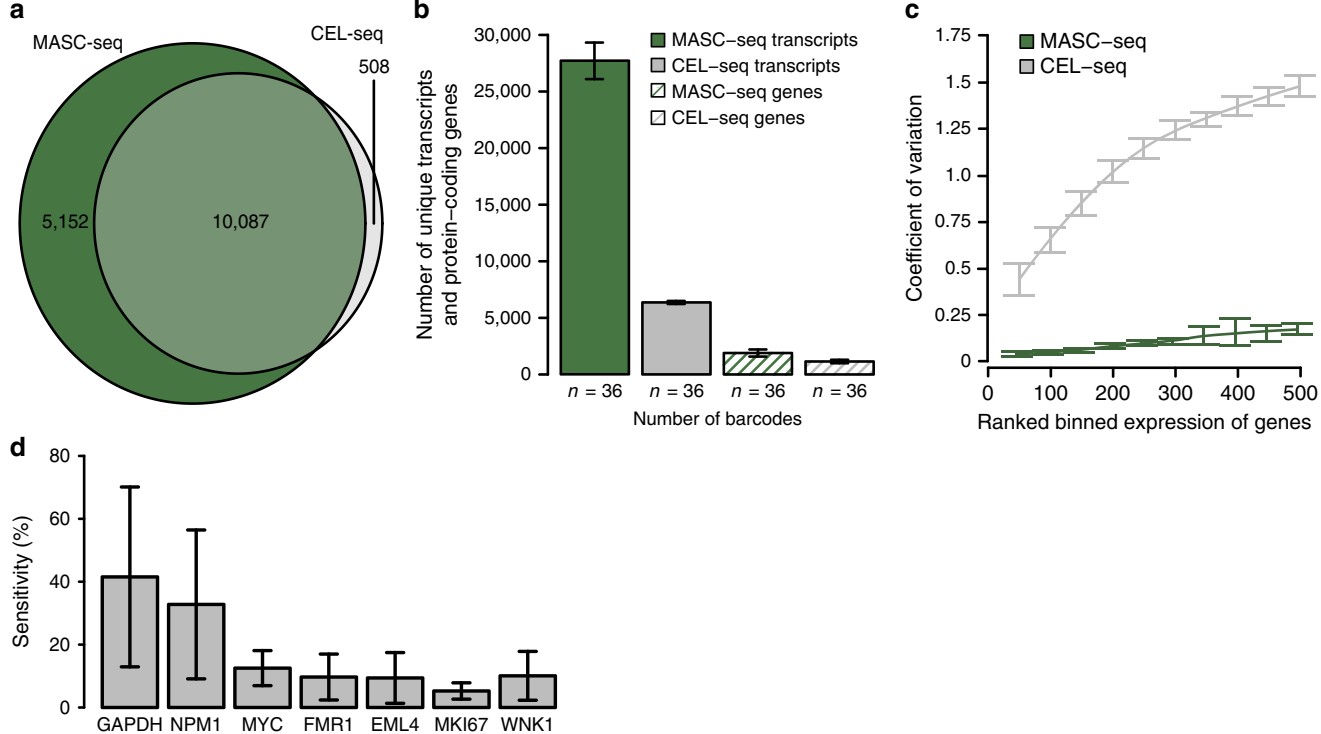

**Figure 4 | Efficiency and sensitivity of MASC-seq.** (**a**) Venn diagram comparing total number of protein-coding genes found in randomly selected single-cell libraries ($n = 36$) created with the MASC-seq and CEL-Seq single-cell approaches ($n = 36$). (**b**) Barplot depicting number of unique protein-coding genes and transcripts found per single cell by each approach. (**c**) CV over ranked gene expression for both techniques. (**d**) MASC-seq sensitivity represented by the percentage of transcripts counted per single cell as compared with smFISH. Errors bars correspond to s.e.m.

Further analysis through $t$-SNE and hierarchical clustering revealed subtle differences between single cells within each CLL subset. A number of major and minor clusters were observed in connection to each subset (Fig. 5a–c).

Differential expression analysis based on the hierarchical clustering results revealed unique expression signatures for each of the clusters. For example, in subset #1, the two minor clusters were defined by strong expression of a number of distinct genes, downregulated in the major cluster (Fig. 5a, Supplementary Fig. 5e). Subset #2 exhibited the strongest expression levels per cell (Supplementary Fig. 5a), possibly related to the proliferative drive and very poor prognosis for patients assigned to this subset[26,27], with all of the clusters in the subset displaying variable expression of *DAB1* and *DAB1-AS1*, which are reportedly involved in cancer progression through NOTCH signalling[28]. The two largest clusters (indicated in black and red in Fig. 5b and Supplementary Fig. 5f) observed in subset #2 had very similar expression patterns, with *C5orf63* being the only overexpressed gene unique for the major cluster (black) while the other cluster (red) exhibited dysregulation in expression of genes like CAMK1 and GLIPR1, connected to renewal of leukaemic stem cells and tumour progression[29,30] (Fig. 5b, Supplementary Fig. 5f). Gene expression levels were notably lower in cells of subset #4 (Supplementary Fig. 5a), a prototype for indolent CLL[26,27], than in cells of the other two subsets. In contrast to the minor clusters, the major cluster of subset #4 were the only cells expressing *DNMT3A* and *EPC1* (Fig. 5c, Supplementary Fig. 5g), both of which are known to be deregulated in acute myeloid leukaemia[31,32]. A complete list of differentially expressed genes is provided in Supplementary Data 1–3.

To further investigate the CLL subsets, we calculated a cell-cycle-specific score for each single cell in each of the subsets (Fig. 5d). Although most of the CLL B cells did not appear to be cycling, as previously reported[33–35], almost 10% of subset #2

cells exhibited strong cell cycle signatures, mostly indicative of commitment to genome replication, in accordance with previous microarray analysis[36]. For all subsets, the cycling cells were contained within the major cluster.

Hierarchical clustering revealed that, as expected, most single cells within each subset had similar expression profiles, and clustered with other cells of their subset, but a few cells from both subsets #2 and #4 clustered with the poor-prognosis subset #1 cells (Fig. 5e, color-coded by subset and by cluster). This cluster containing cells from all three subsets was marked by a differential expression signature of *inhibin beta A* (*INHBA*) (Fig. 5f), a gene associated with cancer progression[37] and poor survival[38].

## Discussion

The importance of heterogeneity between, and within, tumours for therapeutic responses and patient outcomes is well known. However, detecting unique markers within cells of a tumour, predicting associated phenotypic changes and linking the heterogeneity to disease progression is far from straightforward, requiring single-cell analysis[39]. Previous advances in single-cell analysis have enabled measurement of gene expression in a large number of individual cells. This has addressed the drawback of population-averaging in bulk analysis. Further enabling rapid and simple analysis of thousands of cells would help accelerate experimental throughput in both academic and clinical research, potentially resulting in improved patient monitoring, particularly in relation to given therapy.

The presented MASC-seq method for analysing expression profiles in single cells by combining cell imaging with high-throughput single-cell RNA sequencing has been thoroughly validated using both human and mouse cell lines, and primary patient samples. Experiments to evaluate bulk RNA-sequencing

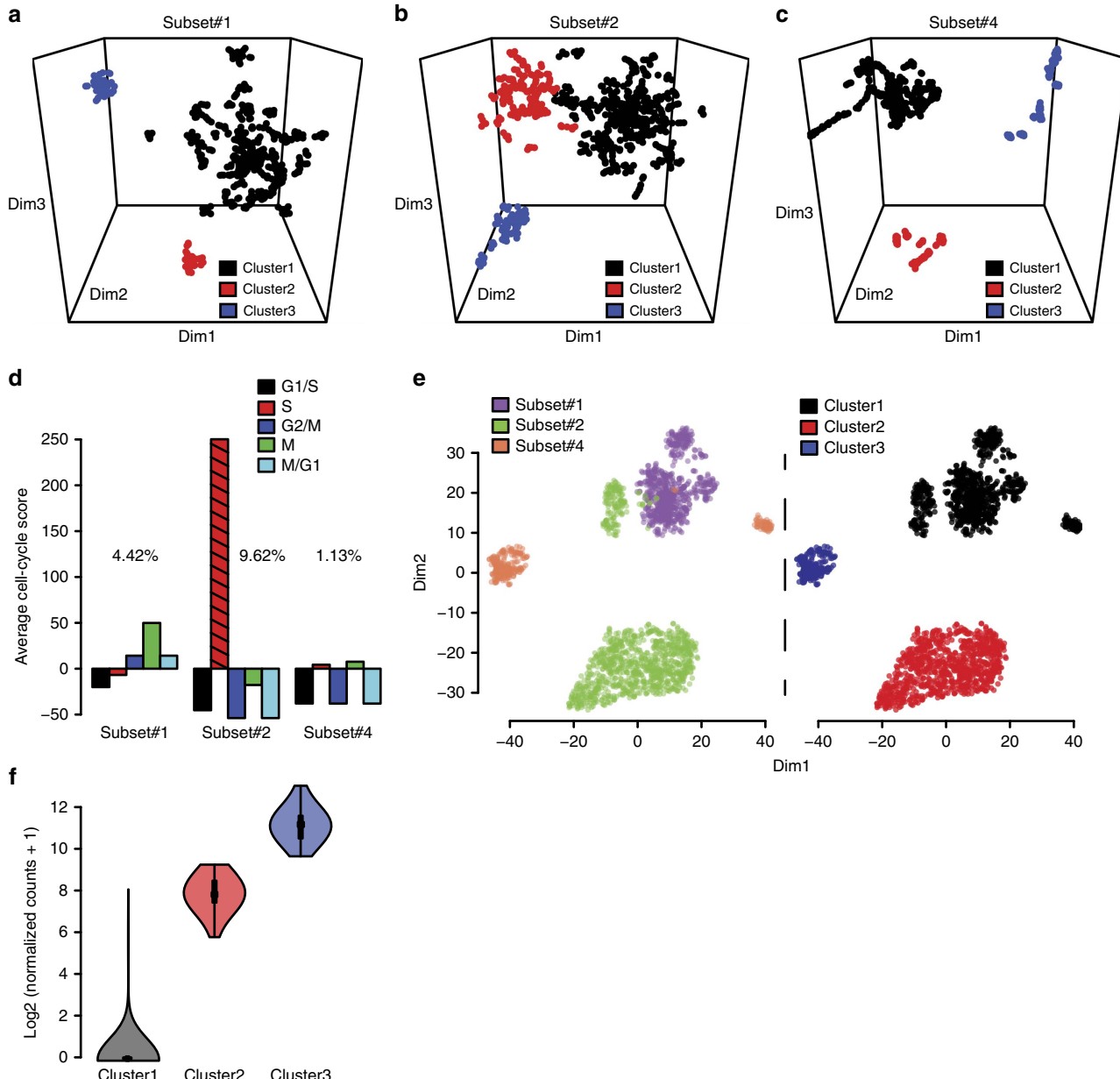

**Figure 5 | Results of analysis of differential expression in single leukaemic cells.** (**a–c**) Clusters in subsets #1, #2 and #4 (respectively) detected by *t*-SNE and hierarchical clustering displayed in three-dimensional plots. (**d**) Barplot of single CLL B cells ordered by average cell cycle score and subset. (**e**) *t*-SNE plot of single CLL cells expressing most abundant genes found in each patient subset (colour-coded by subset in the left panel and by cluster in the right panel). (**f**) Violin plot *INHBA* expression for all three clusters.

CLL data have allowed characterization of CLL patients into subsets[40], but averaging gene expression cannot reveal intra-tumour heterogeneity. Using MASC-seq we observed differences between and within different CLL patient subsets. Also, in each of the subsets, we found a major clone that supports the idea of clonal cooperation and long-lasting clonal equilibrium encouraging overall cancer progression in CLL[41]. Whether these differences are related to the functional pathology of CLL requires further investigation and more extensive studies. Nevertheless, these findings illustrate the heterogeneity within each patient and underscore the importance of analysing transcriptomes at the single-cell level.

The greatest advantage of MACS-seq is generating hundreds of single-cell expression profiles in a single reaction, thus simultaneously lowering technical variability, costs and labour. It can generate expression data from ~10,000 single cells in only 2 days at a cost of just USD 0.13 per cell (Supplementary Data 4), an approach at least 200–300 times lower in cost than the currently most widely applied methods by researchers[10] or commercially[21]. High-throughput approaches such as MASC-seq will greatly facilitate investigations of the biological processes involved in diseases like cancer, and will help to improve our understanding of complex biological phenomena at the single-cell level.

## Methods

**Array production.** Six of the microarrays were printed per Codelink glass slide. Each microarray could be used in an individual experiment, with a specific Illumina indexing primer, and each glass slide could accommodate up to 16 microarrays, but slides with six replicates were prepared and used in the reported experiments to facilitate daily use. The printing process was performed by ArrayJet LTD (Scotland, UK) using the ArrayJet Spider system. The DNA oligonucleotides were spotted in 200 μm centre-to-centre vertical and horizontal pitch fashion. The

arrays' frame was labelled to enable visualization during FACS as described in Supplementary Information.

**Cell handling and total RNA extraction.** MCF-7 (human breast metastatic adenocarcinoma) cells were cultured at 37 °C in a 5% CO$_2$ environment. The breast cancer cells were grown in Eagle's Minimum Essential Medium supplemented with 10% FBS (both from Thermo Fisher Scientific, Life Technologies, Paisley, UK), harvested at 70% plate confluency by trypsinization, and RNA was extracted from ~1 million of the cells using an RNeasy Mini Kit (Qiagen, Limburg, The Netherlands). NIH/3T3 (mouse embryonic fibroblast, hereinafter 3T3) cells were cultured in Dulbecco's Modified Eagle's Medium supplemented with 10% FBS (both from Thermo Fisher Scientific, Life Technologies, Paisley, UK). These 3T3 cells were harvested at 70% plate confluency for RNA extraction, as described above. After trypsinization, the cells were washed twice with 1 × PBS. Total MCF-7 RNA (300 ng) was fragmented to 350 nt fragments, on average, using a magnesium fragmentation protocol[42] involving 5 min incubation at 95 °C and used in all bulk reference experiments, for cDNA synthesis in both solution and on the microarray surface. MCF-7 and 3T3 cells were used in further single-cell experiments by smearing them on the microarray surface after fixation[43]. Before attachment to the array surface, the buffer was exchanged to 0.1 × saline sodium citrate (SSC) and the cells kept on ice until the protocol was started. Smearing was performed by taking 3 µl of cells at a concentration ~2,000 cells µl$^{-1}$ and first slowly pipetting the cells on the array surface with taking care not to actually touch the surface with the tip. Then, the cell solution could be smeared with a side of a pipette tip (again careful not to touch the surface of the array) followed by attachment at 37 °C for 5 min to the slide. MCF-7 was obtained from the German collection of microorganisms and cell cultures (DSMZ) and authenticated with RNA sequencing. The 3T3 cell line was obtained from ATCC. Both cells lines were tested for mycoplasma contamination (Minerva Biolabs Gmbh, Berlin, Germany).

**Patient sample collection.** Cryopreserved peripheral blood mononuclear samples derived from 3 CLL patients were included from the Biobank at Uppsala University Hospital, Sweden. All cases were diagnosed and classified according to recently revised iwCLL criteria[44] with a typical CLL immunophenotype. Cases were selected based on the expression of stereotyped B-cell receptor immunoglobulins from the following major subsets: one subset #1 case (IGHV1/5/7/IGKV1(D)-39 usage), one subset #2 (IGHV3-21/IGLV3-21 usage) and one subset #4 case (IGHV4-34/IGKV2-30 usage). Informed consent was obtained according to the Helsinki declaration and the study was approved by the Regional Ethics Review Committee in Uppsala (2014/33).

**FACS sorting.** The patient samples were collected as described. Before sorting, the cells were fixed in a way to be compatible with staining and immunohistochemical methods[43]. The CD5$^+$CD19$^+$ population (for antibody staining see Supplementary Information) was sorted on the labelled array with position adjustments to increase the sorting accuracy on the correct barcoded DNA oligonucleotide position. The FACS sorter utilized for analyses and single-cell sorting was a BD Influx by Becton Dickinson. Cells analyses and sorting were performed using a 70 µm nozzle. The FACS sorting stage spatial resolution is <100 µm but the droplet size is the one setting the spatial limitations and dictates the sorting strategy. The FACS set-up with the 70 µm nozzle created droplets with a trace diameter of 340 ± 40 µm on the collection slide. To increase the positional precision in single-cell sorting we have used Cell Precision, a kit that integrates an indication laser and a camera beneath the sorting slide to the FACS sorter. The laser trace size on the slide was 330 ± 20 µm and the effective positioning resolution was <100 µm.

The single-cell sorting was performed in successive sorting cycles giving adequate space between neighbouring matrix positions for avoiding droplet fusion. The FACS sorting matrix was 11-by-12 with a matrix unit length of 600 µm (centre-to-centre). The successive sorting cycles differed in the initialization position, which was displaced by 200 µm for every successive cycle at the X dimension (3 cycles) and 200 µm at the Y dimension (3 cycles). In total, we performed 9 sorting cycles to achieve maximum coverage of the oligonucleotides microarrays. Adequate time between successive sorting cycles was given in order for the sorted droplets to evaporate and the encapsulated cell to end up on the oligonucleotides base. The total sorting time per microarray was 8 min. The average efficiency for indexed sorting was 47% (Supplementary Data 5). After sorting, the slide was heated to 37 °C for 5 min.

**Visualization of cell positions.** Images of sorted and stained cells (for staining protocol see Supplementary Information) on barcoded microarrays were recorded using a Metafer Vslide scanning system (MetaSystems, Mannheim, Germany) installed on an Axio Imager Z2 LSM700 microscope (Carl Zeiss, Oberkochen, Germany). All images were taken with the × 20 Plan-Apochromat objective lens, and stitched using the VSlide software (v1.0.0). Before scanning the Cy3 emission range (560–610 nm), the glass slide was mounted with SlowFade Gold Antifade reagent (Life Technologies, Paisley, UK). In case a × 40 objective was used, the scanning was performed on the Zeiss LSM700 system and the images stitched with the ZEN (Zeiss) software. Photoshop CS6 software (Adobe Systems) was used to merge images.

**cDNA synthesis and library preparation.** *Cell permeabilization*. To permeabilize cells on the slide, the slide was placed in an ArrayIT hybridization cassette diving the slide into individual arrays. Then 0.1 × pepsin (Sigma-Aldrich, St Louis, MO) solution (pH 1) warmed to 37 °C was added to each of the sample arrays for 30 s, carefully pipetted out, and the surface carefully washed with 100 µl 0.1 × SSC.

*cDNA synthesis*. The cDNA synthesis mixture contained 1,280 U of Superscript III Reverse Transcriptase, 256 U of RNaseOUT, 3.2 µl of 0.1 M dithiothreitol (DTT) and 0.8 × First Strand buffer (all from Invitrogen, Life Technologies, Paisley, UK), 3.2 µg actinomycin-D (Sigma-Aldrich, St Louis, MO), 0.4 mM dNTPs mix (Thermo Fisher Scientific, Life Technologies, Paisley, UK) and 1.5 × BSA (NEB, Ipswich, MA, USA). The reaction volume was set to 70 µl by adding water. The mix was then added to the microarray surface with the attached ArrayIT hybridization cassette, and then incubated for 16 h. After cDNA synthesis, the microarray surface was washed with 100 µl W3.

*Cell removal*. To remove cells, proteinase K (Qiagen, Limburg, Netherlands) was mixed with proteinase K digestion (PKD) buffer 1:4 then added to the microarray surface with the attached ArrayIT hybridization cassette. After incubation for 1 h at 56 °C, the glass slide was sequentially washed in 2 × SSC supplemented with 0.1% sodium dodecyl sulphate at 50 °C for 600 s, 0.2 × SSC for 60 s and 0.1 × SSC for 60 s at room temperature, then spin-dried.

*Probe release*. A probe release mix was prepared by mixing 6.4 U of USER enzyme and 1.5 × BSA (both from NEB, Ipswich, MA, USA), 0.8 × second strand buffer (Invitrogen, Life Technologies, Paisley, UK), 70 µM dNTP mix (Thermo Fisher Scientific, Life Technologies, Paisley, UK) and water to a final volume of 70 µl, and heated to 37 °C. The glass slide (with the attached ArrayIT hybridization cassette) was then incubated with the mixture for 2 h at 37 °C, then 65 µl of the released probe-containing mixture was collected in a 0.2 ml low-binding tube (Axygen, Corning Life Sciences, Corning, NY). All of the following reactions were performed using low-binding tubes and tips (Biotix, San Diego, CA). Reference material was added as described in Supplementary Information. The array print quality was also examined and used in picture overlay, also described in the Supplementary Information.

*Second strand synthesis and blunting*. Second strands were generated from template cDNA strands using 18.4 U DNA Polymerase I and 0.92 U RNaseH mixed with 0.2 × first strand buffer (all from Invitrogen, Life Technologies, Paisley, UK). The mixtures were incubated for 2 h at 16 °C. To initiate blunting reactions, 15 U of T4 DNA polymerase (NEB, Ipswich, MA) was then added to each mixture for 20 min before the reaction was stopped by adding cold EDTA to a final concentration of 800 µM. Samples were then purified using the Agencourt RNAClean XP system (Beckman Coulter, Pasadena, CA) according to the manufacturer's instructions, and eluted in 20 µl water. Sample volumes were subsequently reduced to ~5.6 µl using a SpeedVac vacuum centrifuge.

*In vitro transcription*. In vitro transcription was performed using a MEGAscript T7 Transcription kit (Ambion, Life Technologies, Paisley, UK) with 1.6 µl of the provided enzyme mix in 1 × reaction buffer and 6.4 µl of the provided nucleoside 5′-triphosphate (NTP) mix, supplemented with 16 U of the SUPERase In RNase Inhibitor (Invitrogen, Life Technologies, Paisley, UK). The mixture was added to the purified product and incubated initially at 37 °C for 14 h and subsequently 4 °C until the sample could be processed. The amplified RNA (aRNA) was then purified using the Agencourt RNAClean XP system (Beckman Coulter) according to the manufacturer's instructions and eluted in 10 µl water. Finally, the purified and aRNA was evaluated using a mRNA Pico Bioanalyzer 2100 system (Agilent, Santa Clara, CA).

*Adaptor ligation and second cDNA synthesis*. After denaturing secondary strands by incubation for 2 min at 70 °C, aRNA adaptors (5′-AGATCGGAAGA GCACACGTCTGAACTCCAGTCAC-3′, 0.7 µM) were ligated to the aRNA strands using 300 U of the truncated T4 Rnl2 (NEB, Ipswich, MA) in 1 × T4 RNA ligase reaction buffer. The adaptors had each been capped at the 3′ end with ddC and adenylated at the 5′ end to mitigate single-strand ligation to the aRNA. 60 U of murine RNase Inhibitor was then added and the mixture was incubated at 25 °C for 1 h. The sample was purified using the Agencourt RNAClean XP system (Beckman Coulter) according to the manufacturer's instructions and eluted in 10 µl water. To generate cDNA, the RT adaptor (5′-GTGACTGGAGTTCAGACGTGTGCTCTTC CGA-3′) was added to a final concentration of 1 µM together with 50 µM dNTPs. To ensure secondary strand denaturation, again, the sample was heated to 65 °C for 2 min, then incubated at 50 °C for 1 h with 200 U SuperScript III Reverse Transcriptase and 40 U RNAseOUT in 1 × first strand buffer and 5 µM DTT. Finally, the acquired cDNA was purified using the Agencourt RNAClean XP system (Beckman Coulter) according to the manufacturer's instructions and eluted in 10 µl water.

*Indexing PCR*. Quantitative PCR was first applied to determine the number of cycles needed for the indexing reaction, using the KAPA HiFi HotStart Ready mix supplemented with 1 × EVA Green, with InPE1.0 (5′-AATGATACGGCGACC ACCGAGATCTACACTCTTTCCCTACACGACGCTCTTCCGATCT-3′) at 5 µM, InPE2.0 (5′-GTGACTGGAGTTCAGACGTGTGCTCTTCCGATCT-3′) at 0.2 µM and the Illumina Indexing primer at 5 µM final concentrations. The indexing reaction was then performed using the same amplification conditions and determined Ct value. The samples were purified using carboxylic acid beads and polyethylene glycol[45], then diluted for sequencing on the NextSeq500 instrument. Reference libraries were constructed as described in Supplementary Information and also sequenced on the NextSeq500 instrument.

*Establishing fluorescent cDNA signatures from single cells.* Cells were routinely attached to the array surface by heating the slide for 5 min at 37 °C, after confirming that they would not detach by placing 121 DAPI-labelled cells on the array (via the FACS procedure), heating, then visually inspecting and counting the cells remaining on the array. The slide was re-examined and the cells were re-counted after washing the glass slide in 2× SSC supplemented with 0.1% sodium dodecyl sulphate at 50 °C for 600 s, 0.2× SSC for 60 s and 0.1× SSC for 60 s at room temperature. Images of the slides were also acquired before and after washing then superimposed. All of the 121 cells remained and did not move on the array surface during the washing steps (data not shown).

To ensure that the cDNA was localized under the cells' surfaces, we performed another straightforward but informative preliminary experiment, in which cells from patient samples or cell lines were attached to the array surface by the FACS procedure or smearing (respectively), stained with haematoxylin, and visualized. The cDNA was then labelled during the cDNA synthesis reaction by supplementing the mixture with 25 μM Cy3-labelled dCTPs (PerkinElmer, Waltham, MA). In addition, the concentration of dCTPs was reduced to 10 μM while concentrations of the remaining dNTPs remained the same. The mixture was incubated overnight at 37 °C, then the cells were removed from the array surface and the underlying Cy3-cDNA print was visualized. Diffusion signatures were estimated using ImageJ. The cells were fluorescently labelled prior as described in the Supplementary information and the cDNA synthesis reaction supplemented as described here. Distance of the Cy3-cDNA signal compared with the cell border was estimated for 9 3T3 cells and 11 MCF-7 cells.

*smFISH.* All probes, except GAPDH, were designed based on our previously described database targeting all human transcripts (www.fusefish.eu/ref. 46)) and consisted of the oligonucleotides listed in Supplementary Data 6. Oligonucleotides with a 3′-TEG amino modification were provided from Biosearch Technologies, and coupled them to Cy5 (GE Healthcare, Cat. Q15108), Alexa Fluor 647 (Molecular Probes, Cat. A37566) or Alexa Fluor 594 (Molecular Probes, Cat. A37565). We fixed cells in methanol-acetic acid 3:1 (v/v) and performed probe hybridization as previously described[46]. We imaged cells at ×100 magnification using high numerical aperture objective (Nikon) on an inverted epifluorescence microscope (Eclipse Ti-E, Nikon) equipped with an EMCCD camera (iXON Ultra 888, ANDOR) and controlled by NIS-Elements software (Nikon). Per field of view, we acquired an image stack consisting of 31 focal planes spaced 0.3 μm apart. We filtered the images and counted mRNA spots using custom-made software written in MATLAB.

**Data analysis.** *Mapping and demultiplexing.* The samples were sequenced in paired end mode. The forward read was sequenced at 31 nt containing the cell barcode sequence and the UMI, and the reverse read consisted of 121 nt providing the matching transcript information. First, the reads were trimmed from both ends on a Burrows-Wheeler aligner (BWA) quality-based approach, and the adaptor sequences were removed. The reads were then mapped to a NCBI's reference human transcriptome using Bowtie2 and annotated against the RefSeq transcriptome reference containing sequences annotated as NM_ and NR_ sequences (as of 11 March 2014). The mapped reads were filtered to identify (and discard) ribosomal sequences.

HTSeq count, with the setting '-intersection-nonempty', was employed to count the number of reads per gene, marking the results as gene expression values in the downstream analysis. The reads could then be demultiplexed, using the 18 nt cell barcode processed with a kmer-approach with two mismatches permitted during the process.

The reads belonging to each of the cell barcodes were subsequently filtered to remove duplicates based on the UMI. To avoid unnecessary delays in data processing, a minimal hamming distance was set and read clusters were created. Potential duplicated sequences were discarded from further analyses. The standard UMI was a semi-randomized 9 nt sequence, WSNNWSNNV, but when analysing reference bulk libraries in solution, it was combined with a fully randomized 8 nt sequence (NNNNNNNN) embedded in the template switch oligo to generate enough possible UMI combinations. UMI-filtered data were used in all of the following data analyses. The reads mapping to gene MALAT1 were removed from analysis due to problems with self-priming. An illustration of data processing is depicted in Supplementary Fig. 6.

*Normalization and data pre-processing.* Expression profiles were normalized by adjusting the total number of UMIs per barcode (corresponding to the total number of transcripts per cell) to 200,000 reads (providing so called TP200K) and adding a pseudocount before transforming the data to log2 scale. In case of CLL samples, data was normalized to 10,000 reads per barcode (so called TP10K).

To assess expression signatures associated with empty barcodes (which did not receive a cell based on the high-resolution image) we first examined expression of the 50 most highly expressed genes apart from recognized 'housekeeping genes'[19], and removed them from the data set. This resulted in removal of 15 genes from the model MCF-7 and 3T3 data sets (Supplementary Data 7). Visual inspection of the reads confirmed that most of them were cytoplasmic non-coding RNA or binding protein sequences, apart from the apparently most abundant mitochondrial signatures. To further investigate these most abundant genes present in the background, we first compared their levels of expression and distributions to those present in the single-cell libraries. All of these highly abundant genes exhibit higher

and more even levels of expression continuously in the single-cell libraries as compared with the background libraries with the expressions of ACTB and GAPDH exemplified in Supplementary Fig. 7a. All the background libraries also correlated well to each other (Supplementary Fig. 7b) concluding the background libraries were similar to each other and most probably a result of background cell-free RNA material present in the cell-suspension buffer before smearing cells onto the array. Similarly, when amount of reads mapping to background barcodes was compared between FACS sorted libraries and smeared libraries, the smeared libraries on average exhibited 11% more reads mapping to the background (Supplementary Fig. 8), at the same sequencing depth, further strengthening the fact the background in depended on the material present in the suspension buffer.

*Expression profiles of single MCF-7 cells.* In experiments which involved smearing MCF-7 cells on the array surface, four sets of libraries were obtained: background (no cells), singles (1 cell), doublets (2 cells) and clusters (>2 cells). Gene expression profiles of 1,021 background, 136 single-cell, 107 doublet and 858 cluster libraries were acquired from three MCF-7 array experiments performed on the same day under the same conditions. Data were filtered based on mean number of reads present in the single-cell libraries, that is, all cell libraries above this threshold were taken into analysis. After data pre-processing, signals from all of the remaining genes were used in subsequent analyses. These included hierarchical clustering of Pearson's correlation distances, and *t*-SNE to visualize the results (although the groups were clearly separated by the first two components). We examined ranked gene expression by linking it to the mean CV for all four groups in the MCF-7 data. We also evaluated cell-to-cell Pearson's correlation coefficients. Finally, DOR were defined as percentage of genes that were not expressed in one data set while showing any levels of expression in the other.

*Barcode cross-talk experiment.* To evaluate barcode crosstalk, 3T3 and MCF-7 cells were mixed in an ∼1:1 ratio, smeared on an array, and expression profiles of barcoded single-cell libraries (identified from high-resolution images, 97 in total) were analysed, as described above. A list of gene orthologs was downloaded from Biomart[47]. Species-specificity was decoded by hierarchical clustering based on non-orthologous sequences (of 2,828 human-specific and 2,348 mouse-specific protein-coding genes) in each of the 97 barcoded libraries. In total 46 MCF-7 and 51 3T3 single cells were correctly identified. We then examined barcode crosstalk by evaluating raw species-specific read counts in each of the demultiplexed single-cell libraries.

We further investigated whether or not the single-cell libraries could be separated using orthologous sequences, by examining cells' correlations based on the 2,000 most variable (based on CV) gene orthologs. *t*-SNE was run for 10,000 iterations on two significant principal component loadings (*P* < 0.01) providing clear species-specific clustering. We also compared the average normalized expression of the decoded species-specific libraries to that of pure species libraries created separately.

*Chronic lymphocytic leukaemia single-cell analysis.* Each patient sample was processed on a separate MASC-seq slide to avoid any chances of cross-contamination between the samples. Expression profiles of 1,102, 1,403 and 1,063 single cells of CLL subset #1, #2 and #4 cells were examined and compared, as follows. The gene expression profiles were normalized (to TP10K values) and transformed to log-scale, as previously described. Sequences of one highly expressed background gene (BCYRN1) were removed (apart from mitochondrial genes) and then genes with average population expression values of at least 6.5 TP10K in each subset were analysed further, following mean-centring of the data. The subsets on average expressed 10,242, 12,228 and 6,782 genes, respectively.

To inspect major differences between the subsets, we examined expression of the 500 most strongly expressed genes in each of the subsets. Subsets #1, #2 and #4 had 469, 474 and 495 abundantly expressed genes not present at high levels in any other subset, respectively. Pearson's correlations between cells expressing these genes were obtained and used in hierarchical clustering and *t*-SNE, the latter based on 47 significant principal component loadings (*P* < 0.01). A likelihood ratio test[48] was used to determine whether the expression patterns of the three clusters determined in the previous step significantly differed (at *P* < 0.001).

Additionally, to evaluate intra-subset differences, we further examined the 2,000 most variable genes (based on CV) in each of the subsets, after reducing dimensions of the distances of Pearson correlation matrices and the significant principal components (*P* < 0.01). *t*-SNE plots were created in three dimensions and hierarchical clustering was applied to each of the clusters identified in the subsets. The significance of differences in expression patterns between the clusters was then tested as above, using the likelihood ratio test and a *P* < 0.001 significance threshold.

*Cell cycle analysis.* A list of cell-cycle-specific genes representing five stages of the cell cycle (G1/S, S, G2/M, M and M/G1) was taken from a published source[49], and genes that passed a certain threshold (*R* > 0.3) in each of the respective cell cycle phases were selected for further analysis. The data were normalized in two steps and average phase-specific score was generated for each cell through all the five phases, to determine its current cell cycle stage[13]. When analysing the CLL samples, a cell-cycle-specific score was further calculated for each patient's subset, thereby creating subset-specific scores.

**Data availability.** The data have been deposited at SRP067878 and at http://www.spatialtranscriptomicsresearch.org/.

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

## Acknowledgements

We thank David Redin for help with illustrations and Marc Friedländer for help with manuscript preparation. We also thank Nicola Crosetto for help with the smFISH data analysis. The data were analysed using resources provided by SNIC through the Uppsala Multidisciplinary Center for Advanced Computational Science (SNIC/UPPMAX) and Bioinformatics Long-Term Support (WABI). This work was supported by the Knut and Alice Wallenberg Foundation, Swedish Cancer Society, Swedish Foundation for Strategic Research, the Swedish Research Council, Tobias Stiftelsen, Torsten Söderbergs Stiftelse, Swedish Research Council Project Research Grant for Junior Researchers (621-2014-5503) and Science for Life Laboratory. We thank the National Genomics Infrastructure (NGI), Sweden for providing infrastructure support.

## Author contributions

S.V. wrote the manuscript and performed experiments and data analysis. P.L.S. helped design the study and guided the experiments. F.S. performed experiments for library construction and designed the fluorescent colocalization protocol. S.G. performed and optimized FACS experiments. J.O.W. performed parts of data analysis. A.M. performed the experiments. J.F.N. developed an alignment and demultiplexing pipeline. J.C. and M.B. performed experiments and analysed smFISH data. L.-A.S. helped with manuscript preparation. R.R. provided clinical samples and helped with manuscript preparation. J.F. helped design the study. J.L. designed the study and guided the experiments and data analysis.

## Additional information

**Competing financial interests:** P.L.S., J.F. and J.L are founders of a company that holds IP rights to the presented technology.

