## [Peer Review File · Nature Communications]

Reviewers' comments:

Reviewer #1 (Remarks to the Author):

This manuscript by Vickovic et al. reports the development of a high throughput microarray based single cell transcriptome platform (MASC-seq). Single cells are positioned on a microarray of barcoded DNA oligos printed in a 33 x 35 matrix (1007 unique DNA barcodes). Images of each cell with barcode are taken and all cells have cDNA RT performed simultaneously. Each single cell microarray has UMI and RTs using oligo-dTVN sequence. The authors use MCF-7 as a model system and "smeared" cells creating four groups of barcoded libraries (single, doublet, cluster, and background - 0 cells). Using a mixture of MCF-7 and 3T3 and species-specific reads, there was approximately 1.5% misassigned barcodes indicating that crosstalk is not significant on this parallel single cell method. They then used three patients with CLL as test cases for their platform.

Comments/Questions

- 1) For clarification, are the printed arrays in individual wells? How does the proximity of one cell with another affect crosstalk? (i.e. if there is a mouse 3T3 cell right next to a MCF-7 human cell, was the crosstalk higher?).
- 2) When one is "smearing" cells on the array, how does one control cell breakage/lysis across multiple arrays? (i.e. is the background - theoretical 0 cells much higher with smearing compared to FACs?)
- 3) As the authors have found consistent with others, proliferation signatures tend to dominate differences in single cell data (Figure 4). Not surprisingly, each subset on their own was quite distinct, but it appears that Cluster 1 is shared by all 3 subsets. What is in Cluster 1? Maybe Figure 4 can be improved by assigning some gene or gene sets to each of the clusters?

Overall, the paper is well written and the technology is similar, but unique to other high throughput single cell methods.

Reviewer #2 (Remarks to the Author):

In this manuscript, Vickovic et al. describe an approach to perform single cell 3' tag mRNA sequencing. It is based on sorting or smearing fixed cells onto spatially barcoded custom microarrays which are labelled with anchored oligodT primers to initiate first strand synthesis. Those primers incorporate said spatial barcode and unique molecular identifier sequences to enable accurate transcript counting. One array contains 1000 spots and can accommodate up to 500 cells for analysis. The authors claim that their method delivers single cell transcript counts for a low price, fast turnaround and low variability across cells. Further, there is an added benefit of being able to image and confirm, eventually even visually examine the isolate cells to provide phenotypical data.

The method is of interest for the wider single cell community if the claims hold true across a range of experiments and laboratories, in particular the promise of being able to image and count transcripts from the same cells. However, there are a few things that need to be addressed in this manuscript to fully convince and convey all the claimed benefits.

Comments:

1. What is the sensitivity of this method? It is similar to CEL-Seq, but does it have similar sensitivity? The authors should provide data, e.g. based on ERCC or SIRV spike-ins, that demonstrates a) ability

to recover a defined number of transcripts (limit of detection) and b) how this number scales with sequencing depth. If this data cannot be provided, the authors should analyse the same cell types with the published CEL-seq method to allow comparison.

2. The authors claim that one benefit is a low variability across cells of one array. How is variability across multiple arrays (on the same slide) and multiple slides processed on different days by different operators?

3. The authors claim a low cost of the method of \$0.13 per cell with no extra equipment required. I find this claim debatable at least and the figure too optimistic. The method is based on Codelink slides, which have a price of \$475 each. Just dividing \$475 by 3000 (500 cells across 6 arrays on 1 slide) gives a cost of \$0.16, without any oligos (about 1000 need to be bought, although they last long) and a variety of enzymes to perform the reactions. Further, the arrays are prepared on an ArrayJet Spider system. This isn't free either, so the author's claim to not require expensive automation (p3 L46) does not hold true here. The Arrayjet company seems to offer a service to print arrays, but that then will drive the price per cell up. In fact, a cheap system like a Formulatrix Mantis (£24,000) or TTP Labtech Mosquito (£40,000) can process thousands of single cell transcriptomes per day and do not require extensive automation knowledge either. Therefore p4 L54 can be omitted. In general, to substantiate the cost benefit claim, it would be interesting to have some supplementary material that details how the authors arrive at \$0.13.

4. The authors use fixed cells and sort (FACS) or smear them onto the glass surface before performing imaging. The images presented in this manuscript are HE-stained cells - is the method compatible with high-resolution (40x, NA >1) fluorescence imaging? Can live cells be maintained and imaged on the slides?

5. There are some mentions of buffers (SSC, W1...W3) that are not clear to me what these are. Please check that all acronyms are introduced properly.

6. It is not clear if the cells are sorted onto a dry array or if there is some form of liquid present on the arrays. If no liquid is present - is that why the cells need to be fixed? If the authors possess a FACS (which one did they use) that has a sufficient spatial resolution to deposit cells within a 100 µm spot, why can one not sort 1000 cells onto the array and fully utilise it? Or is the FACS not precise enough and one needs to put less cells on?

The other aspects of the manuscript seem to be sound, the data analysis follows common procedures and is conclusive. I am no expert in CLL so I cannot judge if the author's findings here. I recommend this manuscript for minor revision.

Reviewer #4 (Remarks to the Author):

In this manuscript, the authors reported a new method for single-cell RNA sequencing, based on capturing single cells onto the spotted DNA clusters on microarray for mRNA binding and reverse transcription. The resulting cDNA molecules that carries cell-specific barcodes were then converted to

double-stranded DNA, in vitro transcribed and eventually converted into Illumina sequencing libraries for sequencing. The major advantage of this method is that each cell can be imaged prior to cell lysis, and the resulting sequencing data can be registered to the morphology. It also has advantage over the Fluidigm C1 platform in term of cost and scalability. The authors demonstrated the utility of this method by sequencing and resolving the heterogeneity among a human adenocarcinoma cell line MCF7, a mouse 3T3 fibroblast line, and human primary single leukemia cells.

Overall, this method appears to be technical sound, and the data analyses were properly handled. It sits somewhere among Fluidigm C1, Drop-Seq, and CytoSeq (Fan et al. Science 2015, should have cited). If the protocol can be reproduced by other lab without excessive debugging, most likely it will have some impacts to the community. However, several aspects need to be improved before this manuscript is publishable.

(1) There is no sufficient description on the details related to how single cells are captured on individual DNA spots on the microarray, how robust is the capture of single cells (by smearing and by sorting), how efficient is the capture of mRNA and cDNA synthesis. Without these critical details, it's hard to judge whether the method can be easily adopted by other groups.

(2) It is important to report some quantitative details on the sequencing data (some were already in Figure S3, but not enough), including total number of sequencing reads per library, fraction of the reads that can be assigned correctly to single cells, mapping rate, clonal rate etc, and compared the performance with DropSeq.

REVIEWERS' COMMENTS:

Reviewer #1 (Remarks to the Author):

The authors have addressed my critiques/questions. The paper is ready for publication.

Reviewer #2 (Remarks to the Author):

I feel that my issues and those raised by the other reviewers have been addressed sufficiently and that the manuscript can be published.

Reviewer #4 (Remarks to the Author):

In this revision the authors have presented the results of additional experiments. The new data demonstrated a good reproducibility of the methods, and also covered a number of important technical details that were missing in the first revision. All my previous concerns have been successfully addressed, and I recommend for the acceptance of this manuscript.

Reviewer #1:

This manuscript by Vickovic et al. reports the development of a high throughput microarray based single cell transcriptome platform (MASC-seq). Single cells are positioned on a microarray of barcoded DNA oligos printed in a 33 x 35 matrix (1007 unique DNA barcodes). Images of each cell with barcode are taken and all cells have cDNA RT performed simultaneously. Each single cell microarray has UMI and RTs using oligo-dTVN sequence. The authors use MCF-7 as a model system and "smeared" cells creating four groups of barcoded libraries (single, doublet, cluster, and background - 0 cells). Using a mixture of MCF-7 and 3T3 and specifies-specific reads, there was approximately 1.5% misassigned barcodes indicating that crosstalk is not significant on this parallel single cell method. They then used three patients with CLL as test cases for their platform.

Comments/Questions

1) For clarification, are the printed arrays in individual wells? How does the proximity of one cell with another affect crosstalk? (i.e. if there is a mouse 3T3 cell right next to a MCF-7 human cell, was the crosstalk higher?).

In line with the suggestion from the reviewers, we have clarified the manuscript further. Arrays are printed to provide multiple sub-arrays (i.e. wells) on a single slide. In our case, six of these sub-arrays are printed on one glass slide (Fig. 1) and reactions are performed in an ArrayIT hybridization cassette providing six separate wells (P 5, L 76 and P 13, L 260-268). Also, to clarify a single well reaction, we have stated in the revised manuscript that the reactions are performed in "a single well" (P 5, L 76).

In order to assess crosstalk/diffusion in our model cell types used in the study (MCF-7 and 3T3), we have analyzed data produced with the cDNA-cell colocalization experiment (Fig. 2). We assessed the spread of the Cy3-cDNA signal outside the cells' borders (with the border defined by either an H&E or fluorescent cell stain). The diffusion was estimated $0.81 \pm 1.46 \mu\text{m}$ for MCF-7 and at $0.86 \pm 1.96 \mu\text{m}$ for 3T3 cells (Supplementary Fig. 1b-c and P 6, L 98-101) indeed demonstrating very limited crosstalk between cells, irrespective of cell species and type.

2) When one is "smearing" cells on the array, how does one control cell breakage/lysis across multiple arrays? (i.e. is the background - theoretical 0 cells much higher with smearing compared to FACs?)

This is a good point brought up by the reviewer. In order to assess the background signals in both approaches, FACS and smearing, we have compared the total percentage of reads mapping to barcodes that have received a cell (as defined with the positional H&E image) and barcodes that have not received a cell (Background). We estimated that the background in case of smearing (MCF-7 cells' reads) is ~11% higher (Supplementary Fig. 8) than FACS-sorted neoplastic B-cell patient samples. The background percentage in both approaches seems very similar irrespective of array/slide it was performed on.

Furthermore, as shown in Supplementary Fig. 1b-c, the labeled cDNA seems to be confined to the cell's borders. Also, when looking at expression of genes like ACTB and GAPDH in the background as compared to the single cell libraries (Supplementary Fig. 7), the background libraries appear as a normal distribution of reads we would expect from a bulk sample and while single cell libraries have a more distinct pattern of ACTB and GAPDH expression per cell (P 23-24, L494-502).

Combined these results lead us to believe that the expression of genes found in the background libraries is a result of cell-free RNA present in the buffer prior to cell lysis as we cannot find concrete evidence of increased cDNA diffusion originating from the cells themselves (after attachment to the array surface).

3) As the authors have found consistent with others, proliferation signatures tend to dominate differences in single cell data (Figure 4). Not surprisingly, each subset on their own was quite distinct, but it appears that Cluster 1 is shared by all 3 subsets. What is in Cluster 1? Maybe Figure 4 can be improved by assigning some gene or gene sets to each of the clusters?

We fully agree with the reviewer and we have included a violin plot (Fig. 5f) marking expression of *inhibin beta A* (INHBA), the only differentially expressed gene in cluster 1 as compared to the remaining clusters in line with the text (P 11, L 221-226).

Overall, the paper is well written and the technology is similar, but unique to other high throughput single cell methods.

We were very happy to read this reviewer's positive comments.

Reviewer #2:

In this manuscript, Vickovic et al. describe an approach to perform single cell 3' tag mRNA sequencing. It is based on sorting or smearing fixed cells onto spatially barcoded custom microarrays which are labelled with anchored oligodT primers to initiate first strand synthesis. Those primers incorporate said spatial barcode and unique molecular identifier sequences to enable accurate transcript counting. One array contains 1000 spots and can accommodate up to 500 cells for analysis. The authors claim that their method delivers single cell transcript counts for a low price, fast turnaround and low variability across cells. Further, there is an added benefit of being able to image and confirm, eventually even visually examine the isolate cells to provide phenotypical data.

The method is of interest for the wider single cell community if the claims hold true across a range of experiments and laboratories, in particular the promise of being able to image and count transcripts from the same cells. However, there are a few things that need to be addressed in this manuscript to fully convince and convey all the claimed benefits.

Comments:

1. What is the sensitivity of this method? It is similar to CEL-Seq, but does it have similar sensitivity? The authors should provide data, e.g. based on ERCC or SIRV spike-ins, that demonstrates a) ability to recover a defined number of transcripts (limit of detection) and b) how this number scales with sequencing depth. If this data cannot be provided, the authors should analyse the same cell types with the published CEL-seq method to allow comparison.

This point is well taken and three series of new experiments have been performed to fully address reproducibility, sensitivity and provide benchmarking. To provide information about sequencing reproducibility we have performed new experiments with the smeared approach on MCF-7 cells. Six arrays in total were processed on a slide - three arrays were processed on one slide by one operator and the same set-up was processed at the same time by a first-time operator on another slide (Fig. 3a-b, Supplementary Fig. 2c-d). No major differences in the quality of generated data were observed between experienced and new user.

Furthermore, as the approach is not intrinsically compatible with using ERCC controls (due to the inability to incorporate ERCC controls in the cells prior to lysis on the array), we have performed additional sets of experiments: firstly, single molecule FISH (smFISH) on a set of seven well characterized genes (expressed at different levels) where we estimated the sensitivity and secondly FACS-sorted single MCF-7 cells into 96-well plates to create CEL-seq libraries (to provide a benchmark).

Sensitivity of MASC-seq, as percentage of transcripts detected per single cell as compared to the smFISH signal, was estimated to 17.3%. With this, MASC-seq surpasses both the classical FACS-sorted CEL-seq approach (12.5%) and the new massive platform Drop-seq (10.7%)

In the benchmark, MASC-seq yielded significantly more data than CEL-seq (on the same cell type) by detecting 1.5x more genes in total, capturing 17.5x more unique transcripts and 6.5x more genes per cell (Fig. 4a-c, and P 8, L 152-161).

The new experiments have been included in the revised version of the manuscript together with more information about mapping, sequencing depth and library diversity, (Fig. 3a-c and P 6, L 107-114.)

2. The authors claim that one benefit is a low variability across cells of one array. How is variability across multiple arrays (on the same slide) and multiple slides processed on different days by different operators?

In line with the response above (point 1), we have repeated the experiments to assess reproducibility (Supplementary Fig. 2c-d). In addition, the variation in gene expression between single cells has been evaluated by comparing the MASC-seq procedure to CEL-seq. As a summary we demonstrate the variability across gene expression is consequently lower with MASC-seq, irrespective of array, slide or operator (Fig. 4c and P 8, L 156-157).

3. The authors claim a low cost of the method of \$0.13 per cell with no extra equipment required. I find this claim debatable at least and the figure too optimistic. The method is based on Codelink slides, which have a price of \$475 each. Just dividing \$475 by 3000 (500 cells across 6 arrays on 1 slide) gives a cost of \$0.16, without any oligos (about 1000 need to be bought, although they last long) and a variety of enzymes to perform the reactions. Further, the arrays are prepared on an ArrayJet Spider system. This isn't free either, so the author's claim to not require expensive automation (p3 L46) does not hold true here. The Arrayjet company seems to offer a service to print arrays, but that then will drive the price per cell up. In fact, a cheap system like a Formulatrix Mantis (£24,000) or

TTP Labtech Mosquito (£40.000) can process thousands of single cell transcriptomes per day and do not require extensive automation knowledge either. Therefore p4 L54 can be omitted. In general, to substantiate the cost benefit claim, it would be interesting to have some supplementary material that details how the authors arrive at \$0.13.

We agree with the reviewer that the cost estimates are important and we believe that the reviewer did not notice that a full cost breakdown was already provided in Supplementary Table 4 in the first submission. This table includes probe synthesis costs, printing services, Codelink slides and reagents for the library preparation (Supplementary Table 4) and referenced to in the text (P 12, L 254). We believe this covers our claim of cost of \$0.13.

However we have modified the text in line with the reviewer suggestions regarding Drop-seq. P 3, L 46 was written in connection to the extensive and expensive instrumentation needed to set-up the Drop-seq protocol. We have now rephrased the statement at P3, L 48 to *“Nevertheless, these methods do not provide any possibilities in combining cell imaging and transcriptome profiling, exhibit low-throughput by analyzing a single cell at a time or require expensive droplet instrumentation when available at high-throughput.”*

4. The authors use fixed cells and sort (FACS) or smear them onto the glass surface before performing imaging. The images presented in this manuscript are HE-stained cells - is the method compatible with high-resolution (40x, NA >1) fluorescence imaging? Can live cells be maintained and imaged on the slides?

We thank the reviewer for bringing up this important issue. The MASC-seq approach is compatible with 40x fluorescence microscopy (Supplementary Fig1a-c and P 5,L 93-94). Live cells cannot be FACS-sorted onto the arrays as they burst upon contact as compared to fixed cells (see image bellow). Currently, live cells unfortunately cannot be maintained on the slide.

5. There are some mentions of buffers (SSC, W1...W3) that are not clear to me what these are. Please check that all acronyms are introduced properly.

We thank the reviewer for these comments. We have now made sure that the SSC acronym is explained at the first mention in the text (P 14, L 285) and the W1-W3 acronyms have been removed.

6. It is not clear if the cells are sorted onto a dry array or if there is some form of liquid present on the arrays. If no liquid is present - is that why the cells need to be fixed? If the authors possess a FACS (which one did they use) that has a sufficient spatial resolution to deposit cells within a 100 μm spot, why can one not sort 1000 cells onto the array and fully utilise it? Or is the FACS not precise enough and one needs to put less cells on?

We apologize for not making clear the detailed information on cell and FACS handling. Cells are sorted dry and are fixed before sorting. Details about the FACS sorting have been added to the Methods part under section FACS sorting (P 15-16, L 304-327) and in the Supplementary Information under section Cell staining for FACS (P 14-15, L 112-129). The sorting accuracy is reported in Supplementary Table 5 and referenced in the text (P 16, L 327).

In short, we have used the BD Influx instrument, which we programed to sort 9 cycles (11x12 cells) per array in a matrix format where we moved the starting position by 200 μm (either horizontally or vertically) for each sorting cycle in order to fill up the theoretical array space. For this, we have developed an array labeling system (Supplementary Information, P 13, L 79-96) and a laser-camera system compatible with the BD Influx machine (P 15-16, L 309-323). As of now, some more improvements would need to be made both to the camera system but also to the BD software

controlling the Influx machine in order to increase the usability of all features on the array.

The other aspects of the manuscript seem to be sound, the data analysis follows common procedures and is conclusive. I am no expert in CLL so I cannot judge if the author's findings here. I recommend this manuscript for minor revision.

We were happy to read this reviewer's supporting comments on our study.

Reviewer #4:

In this manuscript, the authors reported a new method for single-cell RNA sequencing, based on capturing single cells onto the spotted DNA clusters on microarray for mRNA binding and reverse transcription. The resulting cDNA molecules that carries cell-specific barcodes were then converted to double-stranded DNA, in vitro transcribed and eventually converted into Illumina sequencing libraries for sequencing. The major advantage of this method is that each cell can be imaged prior to cell lysis, and the resulting sequencing data can be registered to the morphology. It also has advantage over the Fluidigm C1 platform in term of cost and scalability. The authors demonstrated the utility of this method by sequencing and resolving the heterogeneity among a human adenocarcinoma cell line MCF7, a mouse 3T3 fibroblast line, and human primary single leukemia cells.

Overall, this method appears to be technical sound, and the data analyses were properly handled. It sits somewhere among Fluidigm C1, Drop-Seq, and CytoSeq (Fan et al. Science 2015, should have cited). If the protocol can be reproduced by other lab without excessive debugging, most likely it will have some impacts to the community. However, several aspects need to be improved before this manuscript is publishable.

We fully agree with the reviewer and Fan *et al* has been added as a reference (P 3, L 44). Also, in order to assess reproducibility, robustness and variability between arrays, slides and operators, we have performed a new set of experiments where three arrays were shared by a slide and operator - one operator was an experienced user while the other one was a first time user of the protocol (Fig. 3a-b, Supplementary Fig. 2c-d). Overall, we are happy to find that the presented methodology works well even in the hands of a new user.

(1) There is no sufficient description on the details related to how single cells are captured on individual DNA spots on the microarray, how robust is the capture of single cells (by smearing and by sorting), how efficient is the capture of mRNA and cDNA synthesis. Without these critical details, it's hard to judge whether the method can be easily adopted by other groups.

We agree that a more full description of the details is warranted and we have expanded the text for cell handling and smearing to the array to the Cell handling and total RNA extraction section (P 13-14, L 270-292). Also, more details have been added concerning FACS under the FACS sorting section (P 15-16, L 304-327). Also, information about the sorting accuracy can be found in Supplementary Table 5.

To provide info about sequencing depth and reproducibility we have performed new experiments with the smeared approach on MCF-7 cells. Six arrays in total were processed on a slide - three arrays were processed on one slide by one operator and another three arrays was processed at the same time by a first-time operator (Fig. 3a-b, Supplementary Fig. 2c-d). No major differences were observed.

Furthermore, as the approach is not intrinsically compatible with using ERCC controls (due to the inability to incorporate ERCC controls in the cells prior to lysis on the array), we have performed additional sets of experiments: first single molecule FISH (smFISH) on a set of three known breast cancer genes (expressed at different levels) where we estimated the sensitivity and secondly FACS-sorted single MCF-7 cells into 96-well plates to create CEL-seq libraries to provide a benchmark.

Sensitivity of MASC-seq, as percentage of transcripts detected per single cell as compared to the smFISH signal, was estimated to 17.3%. With this, MASC-seq surpasses both the classical FACS-sorted CEL-seq approach (12.5%) and the new massive platform Drop-seq (10.7%)

In the benchmark, MASC-seq yielded significantly more data than CEL-seq (on the same cell type) by detecting 1.5x more genes in total, capturing 17.5x more unique transcripts and 6.5x more genes per cell (Fig. 4a-c, and P 8, L 152-161).

The new experiments have been included in the revised version of the manuscript supporting the overall performance of the technology (Fig. 3a-c and P 6, L 107-114.)

(2) It is important to report some quantitative details on the sequencing data (some were already in Figure S3, but not enough), including total number of sequencing reads per library, fraction of the reads that can be assigned correctly to single cells, mapping rate, clonal rate etc, and compared the performance with DropSeq.

We agree with the referee and we have in the revised manuscript provided more information about mapping, sequencing depth and library diversity, shown in Fig. 3a-c and P 6, L 107-114.

We estimate that MASC-seq is approximately double the Drop-seq price per cell, however, as no detailed cost estimates were provided in the Drop-seq paper, we chose not to include the direct comparison in our manuscript and instead compared MASC-seq to the most widely used system Fluidigm C1.

In terms of performance, Drop-seq sensitivity is estimated at 10.7% whereas we show that MASC-seq has a sensitivity of 17.3%. Drop-seq's highest purity is reported at 1.2% and we show that MASC-seq at purity of 1.4%. However, both approaches are dependant on the cell concentration load and could be adjusted accordingly. In terms of expression diversity per cell, Drop-seq reported ~44k unique transcripts per one HEK cell while we report ~27k transcripts per single MCF-7 cell. Again, as we did not perform the study on the same cell type these numbers are not fully comparable and we chose not to include such a comparison.